# Using a Dielectrophoretic Microfluidic Biochip Enhanced Fertilization of Mouse Embryo in Vitro

**DOI:** 10.3390/mi11080714

**Published:** 2020-07-23

**Authors:** Hong-Yuan Huang, Wei-Lun Kao, Yi-Wen Wang, Da-Jeng Yao

**Affiliations:** 1Department of Obstetrics and Gynecology, Chang Gung Memorial Hospital, Tao-Yuan 33301, Taiwan; hongyuan@cgmh.org.tw (H.-Y.H.); iwen0711@gmail.com (Y.-W.W.); 2Department of Obstetrics and Gynecology, Chang Gung University and College of Medicine, Tao-Yuan 33305, Taiwan; 3Department of Power Mechanical Engineering, National Tsing Hua University, Hsinchu 30013, Taiwan; kao_wei_lun@hotmail.com; 4Institute of Nanoengineering and Microsystems, National Tsing Hua University, Hsinchu 30013, Taiwan

**Keywords:** oocyte, sperm, dielectrophoresis, single-cell, fertilization in vitro

## Abstract

Droplet microfluidics has appealed to many interests for its capability to epitomize cells in a microscale environment and it is also a forceful technique for high-throughput single-cell epitomization. A dielectrophoretic microfluidic system imitates the oviduct of mammals with a microchannel to achieve fertilization in vitro (IVF) of an imprinting control-region (ICR) mouse. We applied a microfluidic chip and a positive dielectrophoretic (p-DEP) force to capture and to screen the sperm for the purpose of manipulating the oocyte. The p-DEP responses of the oocyte and sperm were exhibited under applied bias conditions (waveform AC 10 V_pp_, 1 MHz) for trapping 1 min. The insemination concentration of sperm nearby the oocyte was increased to enhance the probability of natural fertilization through the p-DEP force trapping. A simulation tool (CFDRC-ACE+) was used to simulate and to analyze the distribution of the electric field. The DEP microfluidic devices were fabricated using poly (dimethylsiloxane) (PDMS) and ITO (indium tin oxide)-glass with electrodes. We discuss the requirement of sperm in a DEP microfluidic chip at varied concentrations to enhance the future rate of fertilization in vitro for an oligozoospermia patient. The result indicates that the rate of fertility in our device is 17.2 ± 7.5% (*n* = 30) at about 3000 sperms, compatible with traditional droplet-based IVF, which is 14.2 ± 7.5% (*n* = 28).

## 1. Introduction

A biochip combining microelectromechanical systems (MEMS) and biomedical technologies has the advantages of biocompatibility, high precision, and processability. The miniaturization of MEMS systems and the rapid detecting particles on the microscale of biological samples make it possible to carry out a small amount of reactions simultaneously. The droplet microfluidic technology has attracted a lot of interest because of its ability to miniaturize cells and many reagents in the microscopic environment. It is also an effective technique for high-throughput single-cell miniatures [1]. With this progress, reproductive medicine has become an important part of the medical field. According to WHO statistics, about 10% to 15% of couples in the world have infertility. Among these, 15% are due to reproductive diseases, 10% to 30% to male fertility disorders, 30% to 40% to female fertility disorders and another 15% to 30% to two fertility disorders. In vitro fertilization (IVF) is a typical assisted reproductive technology (ART) method, involving co-incubation of the sperm and oocytes in vitro to fertilize naturally with an appropriate sperm concentration. The cultured developing embryo is then transferred back to the female uterus. Intracytoplasmic sperm injection (ICSI) is another fertilization technique for severe male factor. During ICSI, a single sperm can be injected directly into an oocyte to form an embryo, but after injection it can damage up to 7% of the oocyte.

For sperm separation, an electrophoresis system for separating human sperm according to size and electric field is proposed. The benefit of this method is that it is fast and safe without the risk of DNA damage [2,3,4,5]. Some research groups use density gradient centrifugation, where selected sperm have discrete density gradients based on the self-promoting activity of sperm. For the oocyte capture section, it has been demonstrated that the dielectrophoresis method manipulates and selects oocytes when applying a waveform (AC 3 V/1 MHz). That result showed that a selected group of oocytes had a developmental potential better than the control group in IVF [6]. Using the dielectrophoretic microfluidic device, the fertilization rate increased to exceed the level of DEP treatment in traditional IVF, and more embryos developed to the blastocyst stage, with a low sperm/oocyte ratio [7]. Recently, the design and manufacture of a planar chip that can be used for high-throughput cell capture and pairing via p-DEP within minutes was completed [8]. Regarding embryo culture, a previous study mentioned that they can enhance the rates of blastocyst development, implantation, and continuing pregnancy of mice using a dynamic-microchannel pulsatile-culture environment [9]. The required sperm concentration within the microfluidic system is another important aspect of fertilization. A droplet-based IVF is normally achievable with a ratio between sperm and oocyte in the range 12,000 to 39,000 [10,11,12]. A droplet of a tiny volume (1–5 μL) can also achieve mouse fertilization in vitro at a sperm–oocyte ratio in a range of 250–350 [13]. Suh et al. discussed the lower limits of total sperm number and sperm concentrations within microchannels [14]; their results indicated that the fertility rate in their chips (27%) is significantly greater than with center-well dishes (10%) for a sperm concentration range from 2 × 10^4^ to 8 × 10^4^ sperm/mL. If IVF is achievable in a microchannel with a smaller number of sperm, the technique becomes applicable to aid an oligozoospermia patient.

Here, our goal is to develop a microfluidic system to mimic the Fallopian tube for insemination, fertilization, and further embryo culture [15,16,17,18]. We hence present a dielectrophoretic chip with ITO-glass electrodes that can serve to capture and to screen the sperm for the purpose of manipulating the oocyte of the ICR (Institute of Cancer Research) mouse, and also to concentrate the sperm at the same locations for embryo formation. We also tested whether a poorly conducting DEP buffer solution is usable for mouse fertilization in vitro, and we compared the rate of fertilization in vitro between a control group and an experimental group at varied concentrations of sperm.

## 2. Materials and Methods

### 2.1. Preparation of ICR Mouse Oocytes and Sperm

The ICR mice were treated according to protocols approved by the Animal Technology Laboratories of Agricultural Technology Research Institute (ATRI-ATL).

(a) Oocyte collection

Briefly, the females were superovulated with an intraperitoneal injection gonadotropin (PMSG, 5 IU), followed by trigger ovulation hCG (5 IU) 42–48 h later. The superovulated mice were sacrificed and the oocytes were obtained on flushing the oviducts after 10–13 h. The oocytes were pre-treated with a micropipette and cultured in a KSOM-AA (potassium simplex optimized medium-amino acid) solution. In some cases, HTF (human tubal fluid) medium was also used as a buffer solution.

(b) Sperm collection

The sperm were obtained from a male mouse (age 6 weeks). The males were euthanized with cranial/cervical dislocation. The cauda portion of the epididymis was minced with scissors and put into KSOM-AA, HTF, and DEP buffers (9.5% sucrose (S7903, Sigma-Aldrich, Saint Louis, MT, USA), 0.1 mg/mL dextrose (D9559, Sigma-Aldrich), 0.1% pluronic F68 (Pluronic F68 non-ionic surfactant 100×, Gibco, Madrid, Spain), respectively. DEP is the phenomenon that involves dielectric forces in a non-uniform electric field. All particles have electrophoretic phenomena in the electric-field environment; the magnitude of the force depends on the electrical properties, shape, and size of the particles, and the rate of change of the field strength. Here we use varied percentages of KSOM-AA to adjust the conductivity of the DEP buffer.

After allowing the sperm to swim out for about 1 h, the sperm used for insemination were collected and incubated at 37 °C in the atmosphere (5% of CO_2_ in the air). The sperm concentration was evaluated using the Makler cell counting chamber. The concentration of sperm in all experiments was about 1.5 × 10^6^ sperm /mL, corresponding to 15,000 sperm in a 10 µL droplet. We applied the same total quantity of sperm with serial dilution within the inlet of the dielectrophoretic microfluidic biochip.

### 2.2. Design and Fabrication of a Microfluidic Biochip for Fertilization in Vitro and an Observation Platform

The fabrication of our DEP microfluidic biochip involved three procedures, first to form the ITO-glass chip with electrodes, second to form the PDMS substrate with a microchannel and micro-structures from a SU8-3050 mold, and third to bond them together with an oxygen plasma, as shown in Figure 1A. The actual device has size 5 cm × 3 cm. Two punched holes (diameter 3.0 mm) served as the inlet and outlet of the reservoir. The outlet reservoir was connected to a syringe pump to control the rate of volume flow. The total volume was about 30 μL in our microfluidic channel. The concept of our experiment was that the mouse oocytes and sperm become trapped on the electrodes for about 1 min with the p-DEP, shown in Figure 1B, with further flow to be co-incubated 1 h at the micro-structures for natural insemination, shown in Figure 1C.

(a) ITO-glass electrode chip

The electrode design for DEP activation is shown in Figure 2. The width of the electrode was 150 μm; the gap between the electrodes was 100 μm. A three-pair electrode design was used to ensure that the oocytes were trapped and fixed, to prevent the loss of oocyte capture at the first two set of electrodes.

We initially cleaned the ITO glass by rigorously washing it three times with acetone, isopropyl alcohol (IPA), and deionized water. HMDS (hexamethyldisilazane) vapor was deposited on the wafer for 5–10 min to increase the adhesion between the spin-coated positive photoresist (AZ5214) (3000 rpm, 30 s) and the ITO glass surface. In order to remove the photoresist solvent, we set the temperature of the soft bake accurately to 100 °C for 1 min. After UV exposure, the photoresist was developed using AZ 400K (AZ 400K: deionized water = 1:5). After patterning the electrodes, the electrodes on the ITO glass substrate were etched at 45 to 46 °C for 40 s with aqua regia (deionized water: nitric acid: hydrochloric acid = 1:0.08:1). To complete the ITO glass electrode chip, ALEG-310 was heated to 60 °C for 5 min to remove the remaining photoresist.

(b) SU8-3050 mold for the PDMS substrate with a microchannel and micro-structures

We fabricated the SU8-3050 mold with a soft lithographic technique. We initially cleaned the glass substrate through rigorous immersion in a piranha solution (sulfuric acid: hydrogen peroxide = 7:1) for 10 min at 90 °C.

The negative photoresist (SU8-3050) was spin-coated at two rates: 500 rpm for 10 s and 1000 rpm for 30 s, respectively. The purpose of the first rate was to make the photoresist completely cover the silicon wafer, and the second rate was to determine the final thickness of the photoresist. The thickness of the SU8 3050 mold was about 140 μm. In order to slowly remove the photoresist solvent, we performed soft baking under heating, raising it by 5 °C every 3 min until it reached 65 °C, and then adjusted to 95 °C after 30 min and waiting for 1 h complete the process. After UV exposure, the photoresist was developed using SU8 developer. When the SU8 structure was completed, the SU8 mold was used for standard cleaning (acetone, isopropanol, and deionized water) to clean the silicon wafer. We poured the PDMS (A: B = 10 g:1 g) into the SU8 mold and cured in a vacuum chamber to remove air bubbles at 85 °C for about 30 min.

(c) Bonding

The PDMS microchannel structure and ITO-glass electrode chip were bonded with an oxygen plasma to complete the DEP microfluidic IVF biochip [19]. The inlet and outlet reservoirs were formed on punching two holes (diameter 3.0 mm) in the PDMS structure. The size of the DEP microfluidic biochip was 5 cm × 3 cm, shown in Figure 1A.

(d) Observation platform

The experiment was observed with an optical microscope (Olympus BX51, Olympus Corporation, Shinjuku-ku, Tokyo, Japan). The electrode pads of the DEP microfluidic chip were connected into a function generator (Agilent, 33220A, Santa Clara, CA, USA) to regulate the AC voltage and frequency. The outlet reservoir was connected to a syringe pump (KD Scientific Syringe Pumps, KDS 220, Holliston, MA, USA) to apply a steady flow field and to control the flow rate, as Hickman et al. indicated that the flow rate and the method of flow delivery might be important factors to affect the embryo development [20]. The CO_2_ incubator (NUAIRE, NU-5500, Plymouth, MN, USA) provided a suitable environment (5% CO_2_, 37 °C) for mouse embryo culture in vitro.

### 2.3. AC Dielectrophoresis

We used commercial CFD software (CFD-RC, CFD-ACE+) to simulate the electric field of our DEP microfluidic chip according to our previous study [18]. The non-uniform electric fields were able to induce a dielectrophoretic force for the manipulation and positioning of the oocytes and sperm. The distribution of electric field around electrodes was determined by CFDRC-ACE + numerical results in the curve graph of the distribution and of the gradient of the square of electric field. The maximum value of the electric field was 66.7 × 10^4^ kV/m, when AC voltage (10 V_pp_ at 1 MHz) was applied, as the oocytes and sperm became trapped and positioned with an effective electric field under a positive DEP regime. Using 10 V_pp_ 1 MHz can also avoid parthenogenesis [21]. The microchannel had height 140 μm and width 520 μm. The gap between two electrodes (width 150 μm) was 100 μm.

The sources of the dielectric properties of cells were the sperm and oocyte itself. The gamete cells (oocyte and sperm) were electrogenic, capable of responding to electrical stimuli and modifying their electrical properties during the crucial periods of maturation and fertilization. Furthermore, ion channels have been widely demonstrated on the plasma membrane of the oocyte and spermatozoon in all animals studied, and electrical modifications in gametes are due to ion currents that are modulated via these ion channels. The modification of intracellular calcium levels in gametes has been recognized to be a second messenger system for gamete maturation and fertilization [22,23,24,25].

Assuming DEP buffer solution, the relative dielectric constants of oocytes and live sperm were 78.5, 70, and 50, respectively. The radius of mouse oocytes and sperm were 50 and 5 μm, respectively. The conductivity of the DEP buffer solution was 0.00056 S/m. The conductivity of oocytes and sperm was assumed to be 0.02 and 0.6 S/m. The capacitance of the membrane was 1.25 μF/cm^2^. To confirm that oocytes and sperm will not lyse under the applied electric field, we calculated the transmembrane voltage of oocytes and sperm as spherical cells as follows [21,26]. Typically, for induced transmembrane voltages greater than 100 mV, electroporation occurs on the cell membrane. Avoiding damage to oocytes and sperm by electroporation is very important in our work.

In
(1)$$V_{tm}=\frac{1.5\left|E\right|a}{\sqrt{1+{\left(\omega \tau \right)}^{2}}}$$
the time coefficient to charge the cell membrane is defined as
(2)$$\tau_{mem}=aC_{mem}\left(\frac{1}{\sigma_{cyto}}+\frac{1}{2}\frac{1}{\sigma_{medium}}\right)$$
with cell radius *a*, intensity |E| of electric field and applied frequency ω. Under a condition *E*_max_ = 66.7 kV/m at 1 MHz, $\tau_{mem}$ of oocyte and sperm are 0.59 ms and 55.9 μs; the trans-membrane voltages of the oocyte and sperm were 8.47 mV and 8.95 mV, i.e., less than 100 mV. We hence assumed that the oocyte and sperm are not harmed under our conditions [27].

### 2.4. Experimental Process for Standard IVF as a Control Group

A dielectrophoretic force is commonly applied in cell manipulation [28,29,30]. In assisted reproductive technique (ART), a conventional medium for fertilization in vitro (IVF) and culture in vitro (IVC) such as HTF or KSOM-AA has a large conductivity, which might result in an unexpected heating effect during DEP that would cause damage to the sperm and oocytes. The conductivity of a DEP buffer solution (4.3 μS/cm) was significantly less than that of a conventional IVF medium (>16 mS/cm). To ensure that the low conductivity of DEP buffer can be used for IVF procedures in ICR mice model, we compared the in vitro fertilization rate and blastocyst formation rate of traditional medium, KSOM-AA, HTF and DEP buffer. The experimental process was the same as the standard IVF protocol. A droplet (approximately 30 μL) was loaded into the Petri dish with the desired sperm concentration (1.0 × 10^6^ sperm/mL); then approximately 15–20 oocytes were introduced to each droplet by precision pipetting. Sperm and oocytes were co-cultured for 1 h in a humidified incubator (5% CO_2_, 37 °C). After insemination, all oocytes of these three groups were washed three times with KSOM-AA medium. The zygotes were transferred into another pre-equilibrated fresh KSOM-AA droplet (30 μL) under mineral oil on a plastic Petri dish, and then incubated in a humidified incubator (5% CO_2_, 37 °C). Fertilization was assessed one day after insemination and was defined rigorously as the occurrence of early cleavage through which an embryo developed to a two-cell stage. The embryo development was observed with an optical microscope as a function of period (E1.5, E2.0, E2.5, E3.0, E3.5, E4.0, and E4.5 days). We compared also the rate of fertilization in vitro at sperm concentrations from 3000 to 240,000 per mL in the KSOM-AA droplet (30 μL), which was also incubated in a humidified incubator (5% CO_2_, 37 °C).

### 2.5. Experimental Process Using the Microfluidic Chip

In the manual method the mouse oocytes were fertilized in vitro and cultured as follows. The oocytes were washed three times in a DEP buffer solution to maintain a small conductivity of the working fluid before further experiments. To manipulate the oocytes and sperm with the p-DEP force, we set the conditions of the AC pulse to be 10 V_pp_ at 1 MHz.

We loaded the oocytes and sperm into the inlet reservoir with a pipette. The outlet of the chip was connected into a syringe pump to provide a steady flow; the oocytes were pulled at the same time at 0.5–1.0 μL/min. On applying AC voltage 10 V_pp_ at 1 MHz to generate a strong electric field, the induced p-DEP force trapped oocytes and sperm at the gaps of the two ITO-glass electrodes while they passed through the electrode pads. We tested also the rate of infertility in our DEP microfluidic chip at inseminated sperm numbers varying from 3000 to 240,000. The mouse oocytes and sperm were trapped with the p-DEP force between the gaps of the electrodes for about 1 min and co-incubated for about 1 h for natural insemination inside our microfluidic channel. The zygotes were then removed and washed three times with the KSOM-AA medium. The zygotes were eventually cultured in another pre-equilibrated fresh KSOM-AA droplet (30 μL) under mineral oil on a plastic Petri dish and incubated in a humidified incubator (5% CO_2_, 37 °C) [31,32]. The embryo development was tracked through observation with the optical microscope for periods E1.5, E2.0, E2.5, E3.0, E3.5, E4.0, and E4.5 days.

## 3. Results and Discussion

### 3.1. Standard IVF Control Group

To ensure that the DEP buffer solution had no significantly deleterious effect on the IVF procedure, the experiments included two control groups with HTF or KSOM-AA solutions. For insemination (1 h) at sperm concentration 1.0 × 10^6^ sperm/mL, the rates of fertility and blastocysts were compared between HTF, KSOM-AA, and DEP buffer solutions. Figure 3A shows that the rates of fertilization in vitro were HTF (45.0 ± 4.8%; *n* = 94), KSOM-AA (50.2 ± 6.0%; *n* = 96), and DEP buffer solution (45.1 ± 7.9%; *n* = 37). There was no significant statistical difference between DEP buffer solution and traditional culture medium (*p* > 0.1). The blastocyst rates were HTF (28.6 ± 12.2%; *n* = 26), KSOM-AA (37.4 ± 8.0%; *n* = 34), and DEP buffer solution (30.9 ± 13.2%; *n* = 30), shown in Figure 3B, for which there was also no significant statistical difference (*p* > 0.1). Tracking of the embryo development showed that the embryo developed normally up to the blastocyst stage. Figure 3C indicates that the oocyte and sperm suffered no harm after IVF in the low-conductivity medium.

### 3.2. DEP Microfluidic IVF Chip

Figure 4A shows that the mouse oocytes and sperm were loaded into a microchannel at volume flow rate 0.5–1.0 μL/min generated with the syringe pump connected to the outlet of the chip. Figure 4B shows that, at the applied AC voltage 10 V_pp_ and frequency 1 MHz, the oocytes and sperm were trapped with the p-DEP force between the two electrode pads. The local sperm concentration increased to 1.0 × 10^8^ sperm/mL from the original 1.0 × 10^5^ sperm/mL. The white arrow shows the direction of the flow. Figure 4C indicates that, once the p-DEP force was released after trapping for 1 min, the oocytes and sperm were moved and positioned by the microstructure behind the microchannel for natural insemination for 1 h. Figure 4D indicates that a second polar body with two pronuclei (male and female pronuclei) was observed after 4–10 h. Without this observation of a second polar body, the results likely show a parthenogenesis effect and failure of fertilization in vitro.

Figure 5A shows that the in vitro fertilization rate of DEP microfluidic chip is compared with the traditional IVF under different sperm counts (3000–240,000 sperm). In this study, the rate of fertilization was proportional to the sperm concentration within the range of 18,000–240,000 sperm counts in the DEP microfluidic chip and traditional IVF group. The rate of fertility in vitro in the DEP microfluidic chip non-significantly increased 6.9% at the sperm 9000 count and 21.1% at sperm 3000 count beyond that with the traditional IVF group. At total sperm count 3000, the fertility rate was improved 21.1% in our DEP microfluidic chip. We have demonstrated that even with a small number of sperm, the results show that this designed microfluidic chip is useful to providing an alternative to intracytoplasmic sperm injection for fertilization in patients with severe oligospermia. Figure 5B shows the tracking of embryo development in vitro observed with an optical microscope at periods E1.5, E2.0, E2.5, E3.0, E3.5, E4.0, and E4.5 days, which indicated that the oocyte and sperm suffered no harm after p-DEP trapping and IVF in a low-conductivity medium. The blastocyst rate with the DEP microfluidic chip was 33.6 ± 15.0% (*n* = 26), compared with 37.4 ± 8.0% (*n* = 34) with the traditional droplet-based IVF; there is no significant statistical difference (*p* > 0.1).

## 4. Conclusions

Normally we fertilize the human egg in vitro with a 30,000 to 50,000 sperm in a microdrop culture. This is also the main basis for our development of this chip; because oligozoospermia patients cannot fertilize in traditional IVF programs, fertilization has been carried by sperm injection. We have demonstrated the DEP microfluidic biochip, which can imitate the fallopian tubes and increase the rate of in vitro fertilization of ICR mice. Oocytes and sperm were manipulated and positioned in the microchannel with p-DEP force. When a voltage of 10 V_pp_ was applied at 1 MHz to generate a powerful electric field (66.7 kV/m), the induced p-DEP force trapped mouse oocytes and sperm in a region with high electric field strength. The results of the DEP microfluidic chip show that this capture method can increase sperm concentration. Sperm concentration near oocytes increased from 1.0 × 10^5^ to 100 × 10^6^ sperm/mL in the DEP microfluidic chip. We proved also that a medium of low conductivity caused no damage to the mouse oocyte and sperm. The results indicate that the rate of fertility in vitro in our DEP microfluidic chip is 17.2 ± 7.5% at sperm number 3000 in which was compatible with the traditional rate of IVF 14.2 ± 7.5%. In this work, we imitate the oviduct to position the oocytes with a dielectrophoretic (DEP) microchannel chip technique effectively. The biochip increases the opportunity of sperm in natural fertilization and directly replaces the traditional microinjection. A future goal is to apply this designed microfluidic chip to aid oligozoospermia patients.

## Figures and Tables

**Figure 1 micromachines-11-00714-f001:**
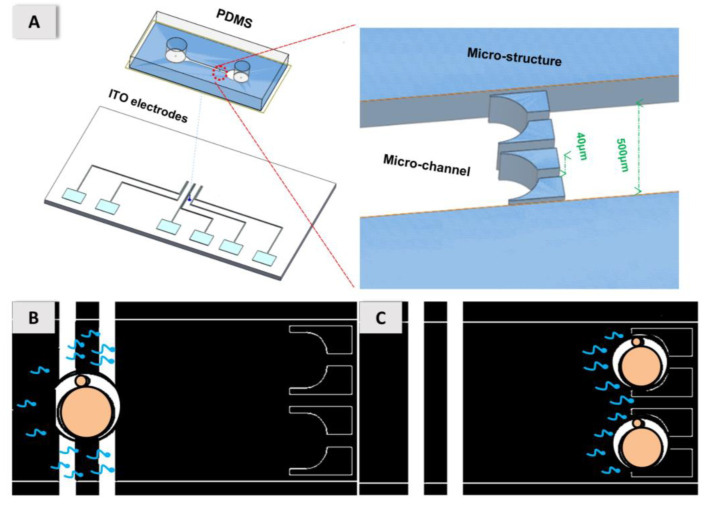
Fabrication of the designed DEP microfluidic biochip; (**A**) actual device of size 5 cm × 3 cm. Bonding of PDMS (microchannel and micro-structures) with the ITO electrode chip. (**B**) The concept of the experiment is that the mouse oocytes and sperm become trapped on the electrodes for about 1 min with the p-DEP. (**C**) Co-incubated 1 h in the micro-structures for natural insemination.

**Figure 2 micromachines-11-00714-f002:**
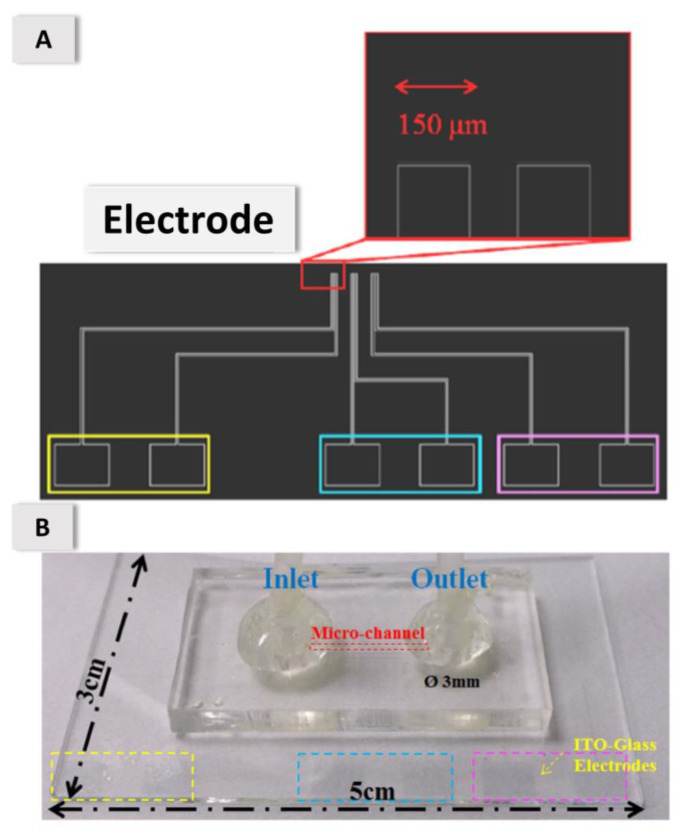
(**A**) Three pairs of parallel linear ITO-glass electrodes. (**B**) Actual device (5 cm × 3 cm). Each gap between the electrodes has width 100 μm; each electrode has width 150 μm.

**Figure 3 micromachines-11-00714-f003:**
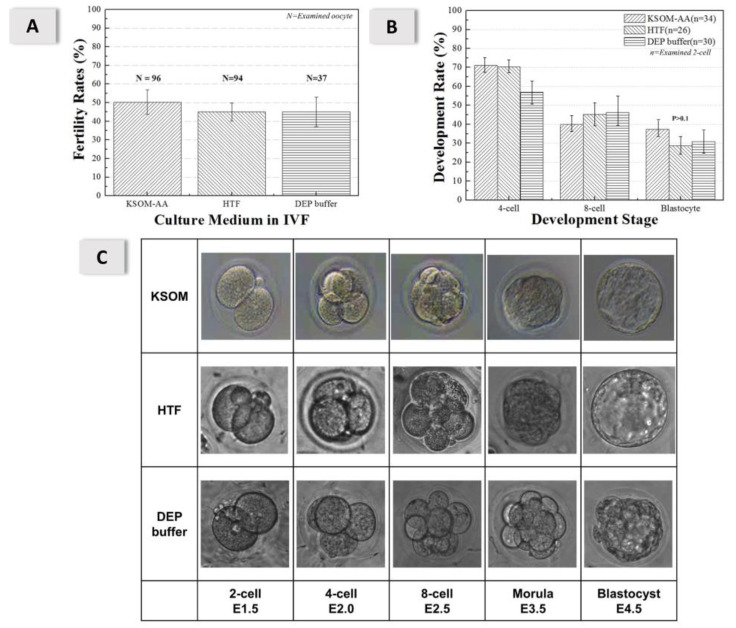
(**A**) Rates of fertilization in vitro under KSOM-AA (50.2 ± 6.6%; *n* = 96), HTF (45.0 ± 4.8%; *n* = 94), and DEP buffer solution (45.1 ± 7.9%; *n* = 37). There is no significantly statistical difference among conditions (*p* > 0.1). (**B**) Blastocyst rates under KSOM-AA (37.4 ± 8.0%; *n* = 34), HTF (28.6 ± 12.2%; *n* = 26), and DEP buffer solution (30.9 ± 13.2%; *n* = 30), respectively; there is no significantly statistical difference (*p* > 0.1). (**C**) Tracking of embryo development of these three groups with an optical microscope at six periods (E1.5, E2.0, E2.5, E3.0, E3.5, E4.0, and E4.5 days); magnification 40×. (All sizes of embryo pictures are about 100 μm 100 μm).

**Figure 4 micromachines-11-00714-f004:**
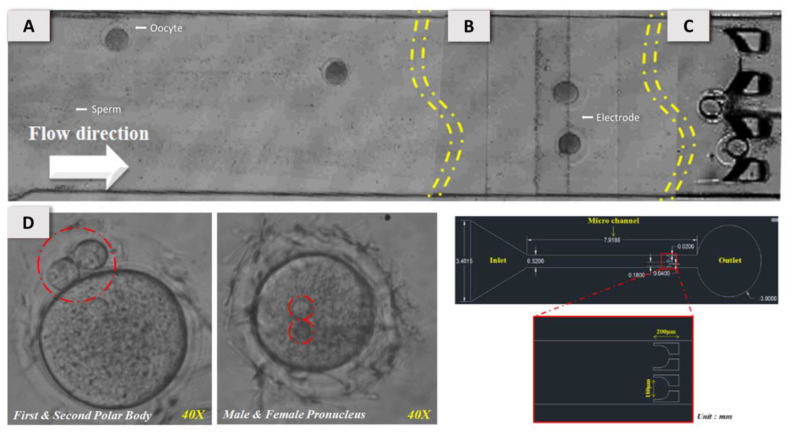
Illustration of the experimental process in the DEP microfluidic chip. (**A**) The mouse oocytes and sperm were loaded into a microchannel with volume flow rate 0.5–1.0 μL·min^−1^. (**B**) At the applied AC voltage 10 V_pp_ at 1 MHz, the oocytes and sperm were trapped with the positive dielectrophoretic force between the two electrode pads. The local sperm concentration increased to 100 × 10^6^ sperm·mL^−1^, regardless of the original sperm concentration. The white arrow shows the direction of the flow. (**C**) On release of the p-DEP force after trapping for 1 min, the oocytes and sperm were trapped with the microstructure behind the microchannel for natural insemination for 1 h. (**D**) A second polar body and two pronuclei were observed after 4–10 h.

**Figure 5 micromachines-11-00714-f005:**
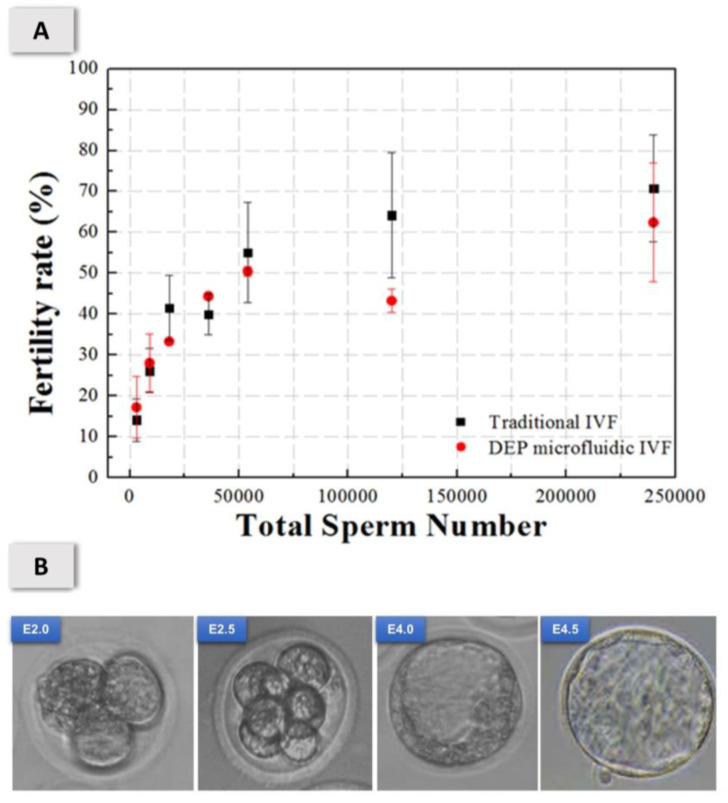
(**A**) Rate of fertilization in vitro of a DFP microfluidic chip compared with traditional IVF at sperm concentrations of 3000–240,000 sperm. (**B**) Tracking of embryo development observed with an optical microscope; magnification 40×. The blastocyst rate was 33.6 ± 15.0% (*n* = 26) of the DEP microfluidic IVF chip, compared with 37.4 ± 8.0% (*n* = 34) of the traditional droplet-based IVF.

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
