# Peer review of "Using a Dielectrophoretic Microfluidic Biochip Enhanced Fertilization of Mouse Embryo in Vitro"

_micromachines, 2020, doi:10.3390/mi11080714_

Round 1
Reviewer 1 Report
The paper provides an interesting method for fertilization. Please justify how this approach is advantageous over using IVF.
Author Response
Comments and Suggestions for Authors
The paper provides an interesting method for fertilization. Please justify how this approach is advantageous over using IVF.
Answer:
Thank you very much for your comments and interesting in this manuscript. Normally we will fertilize the human egg in vitro with a 30, 000 to 50, 000 sperm in microdrop culture. This is also the main basis for our development of this chip, because oligozoospermia patients cannot fertilize in nature traditionally in IVF program, it must be carried by ICSI. Our chip is focused on hoping to replace invasive ICSI here. This design mainly intended to replace the traditional intracytoplasmic sperm microinjection (ICSI) for oligospermia patients to achieve the purpose of egg fertilization in ART program.
We’ve been demonstrated that DEP was used to successfully complete the embryo formation in our previous study; and this study mainly discusses sperms of different concentrations, and can also achieve results with a very low concentration (<10000 sperm) for the clinical application in future. In addition, the traditional IVF technique generally requires medical experts to select appropriate sperm and to fix oocytes with microscopic operation. This process is a serious drain on medical manpower and might damage the quality of the oocytes. In this work, we imitate the oviduct, so effectively to position the oocytes with a dielectrophoretic (DEP) microchannel chip technique. The biochip increases the opportunity of sperm in natural fertilization and directly replaces the traditional microinjection.
We’ve revised and highlight the advantage in revised manuscript.
Reviewer 2 Report
The authors presented a novel in vitro fertilization (IVF) platform which integrates microfluidic channel with DEP technology. The results show that the IVF rate of the new platform is comparable with the conventional IVF approach. Experiments were properly designed to investigate possible harm to cells from DEP solution and exerted electric field. The results of this study will provide some valuable insight to researchers working on IVF. However, some issues should be addressed prior to being considered for publication in the journal Micromachines.
Major issues:
- The electric field magnitude generated by ITO glass electrodes are calculated with CFD software. The authors should provide a figure showing the distribution of electric field around electrodes instead of just providing the maximum value.
- What are the sources of the dielectric properties of cells? References should be provided.
- Why 10 Vpp 1MHz signal are selected for cell capture? The authors should specify whether it is from calculation or experimental test.
- Why three sets of electrodes are created? Is it because three sets can provide sufficient cell capture capability? Please specify.
- How do you determine the local concentration of sperm trapped near electrodes?
Minor issues:
- In line 125, does “Wang aqueous solution” mean aqua regia?
- There are some grammar mistakes and a couple of typos. Please read through carefully and fix them.
Author Response
Comments and Suggestions for Authors
The authors presented a novel in vitro fertilization (IVF) platform which integrates microfluidic channel with DEP technology. The results show that the IVF rate of the new platform is comparable with the conventional IVF approach. Experiments were properly designed to investigate possible harm to cells from DEP solution and exerted electric field. The results of this study will provide some valuable insight to researchers working on IVF. However, some issues should be addressed prior to being considered for publication in the journal Micromachines.
Major issues:
- The electric field magnitude generated by ITO glass electrodes are calculated with CFD software. The authors should provide a figure showing the distribution of electric field around electrodes instead of just providing the maximum value.
Answer:
Thanks for reviewer’s great question. Briefly, we used CFD software (CFD-RC, CFD-ACE+) to simulate the electric field of our DEP microfluidic biochip according to our previous study [18]. The non-uniform electric fields were able to induce a dielectrophoretic force for the manipulation and positioning of the oocytes and sperm. The distribution of electric field around electrodes was determined by CFDRC-ACE+ numerical results in the curve graph of the distribution and of the gradient of the square of electric field. The maximum value of electric field is 66.7 x104kV/m, when AC voltage (10 Vpp at 1 MHz) was applied, as the oocytes and sperm become trapped and positioned with an effective electric field under a positive DEP regime. We’ve revised in the revised manuscript. (Ref#18. Huang H.Y. et al. Biomicrofluidics, 2015, 9:022404., Figure 6)
- What are the sources of the dielectric properties of cells? References should be provided.
Answer:
The sources of the dielectric properties of cells are sperm and oocyte itself. The Gamete cells (oocyte and sperm) are electrogenic, capable of responding to electrical stimuli and modifying their electrical properties during the crucial periods of maturation and fertilization. Furthermore, ion channels have been widely demonstrated on the plasma membrane of the oocyte and spermatozoon in all animals studied, and electrical modifications in gametes are due to ion currents that are modulated via these ion channels. The modification of intracellular calcium levels in gametes has been recognized to be a second messenger system for gamete maturation and fertilization. We’ve revised and referenced additional reference in revised manuscript. (Ref #22-25)
- Why 10 Vpp 1MHz signal are selected for cell capture? The authors should specify whether it is from calculation or experimental test.
Answer:
10 Vpp 1MHz signal are selected for cell capture as well as to prevent parthenogenesis phenomena according to our earlier study (Ref# 21), The purpose of using 10 Vpp 1MHz can also avoid parthenogenesis. We’ve revised in the revised manuscript. (Ref#21. Huang H.Y. et al. Micro and Nano letters, 13:794-797, 2018., Figure 6)
- Why three sets of electrodes are created? Is it because three sets can provide sufficient cell capture capability? Please specify.
Answer:
We used three set of electrodes to prevent the loss of oocyte capture at the first two set of electrodes.
- How do you determine the local concentration of sperm trapped near electrodes?
Answer:
The sperm for insemination were collected and incubated. The concentration of sperm in all experiments was about 1.5´106 sperm /ml, corresponding to 15000 sperm in a 10 ml droplet. We applied the same total quantity of sperm with serial dilution within the inlet of the dielectrophoretic microfluidic biochip. The local concentration of sperm trapped near electrodes was estimated qualitatively microscope direct observation, normally it is around approximately two thousand.
Minor issues:
- In line 125, does “Wang aqueous solution” mean aqua regia?
Answer:
Yes, the aqua regia. We’ve revised in revise manuscript
- There are some grammar mistakes and a couple of typos. Please read through carefully and fix them.
Answer:
Thank you for your suggestion. We have made substantial revisions to our manuscript, both in terms of grammar and content. The revised version of the manuscript has also been edited by a native-English speaking specialist.
This manuscript is a resubmission of an earlier submission. The following is a list of the peer review reports and author responses from that submission.
Round 1
Reviewer 1 Report
The manuscript “Using a Dielectrophoretic Microfluidic Biochip Enhanced Fertilization of Mouse Embryo in Vitro” by Hong-Yuan Huang et al. presents a microfluidic device utilizing DEP for fertilization of Mouse Embryo by concentrating sperm and oocyte. The authors investigate the rates of fertilization using various buffers and showed that there is no significant effect of the DEP buffer on fertilization. Next, the rates of fertilization in the microfluidic device are investigated after trapping sperm and oocyte using p-DEP. The sperm and oocyte were trapped using p-DEP and the fertilization was demonstrated in the microfluidic device.
Although the concept of fertilization in the microfluidc device seems work, what is the main merit of the microfluidic device. The rates of fertilization in the microfluidic is similar with that of conventional method. The process is not automated and complex processes are required to operate the device. Moreover, the concept of trapping sperm and oocyte was already introduced in their previous publication (Huang et al., Biomicrofluidics, 2015, 9:022404). Hence, I recommend rejection of the manuscript.
Additional comments are shown as follows:
- The authors introduced oocyte trapping pockets into the microfluidic channel. Why the pockets are required? The main function and performance of the structure should be addressed clearly.
- The quality of Figure 4 should be improved. Fluorescence image after labeling would help to express their results.
- The trapping efficiency of sperm and oocyte should be addressed.
- The shape of the sperm should be taken into account to estimate transmembrane potential.
Author Response
Reviewer 1:
Comments and Suggestions for Authors
The manuscript “Using a Dielectrophoretic Microfluidic Biochip Enhanced Fertilization of Mouse Embryo in Vitro” by Hong-Yuan Huang et al. presents a microfluidic device utilizing DEP for fertilization of Mouse Embryo by concentrating sperm and oocyte. The authors investigate the rates of fertilization using various buffers and showed that there is no significant effect of the DEP buffer on fertilization. Next, the rates of fertilization in the microfluidic device are investigated after trapping sperm and oocyte using p-DEP. The sperm and oocyte were trapped using p-DEP and the fertilization was demonstrated in the microfluidic device.
- Although the concept of fertilization in the microfluidc device seems work, what is the main merit of the microfluidic device. The rates of fertilization in the microfluidic is similar with that of conventional method. The process is not automated and complex processes are required to operate the device. Moreover, the concept of trapping sperm and oocyte was already introduced in their previous publication (Huang et al., Biomicrofluidics, 2015, 9:022404). Hence, I recommend rejection of the manuscript.
Answer:
Normally we will fertilize the egg in vitro with a 30, 000 to 50, 000 sperm in microdrop culture. This design mainly intended to replace the traditional intracytoplasmic sperm microinjection (ICSI) for oligospermia patients to achieve the purpose of egg fertilization in ART program. We’ve been demonstrated that DEP was used to successfully complete the embryo formation in our previous study; and this paper mainly discusses sperms of different concentrations, and can also achieve results with a very low concentrations (<10000 sperm) for the clinical application in future.
Additional comments are shown as follows:
- The authors introduced oocyte trapping pockets into the microfluidic channel. Why the pockets are required? The main function and performance of the structure should be addressed clearly.
Ans:
The pocket is used for embryo culture design in dielectrophoretic microfluidic biochip. It’s also plays as a micro-structure without voltage application for natural insemination in chip. The experiment was observed with an optical microscope and the electrode pads of the DEP microfluidic chip were connected into a function generator to regulate the AC voltage and frequency. The flow rate and the method of flow delivery might be important factors to affect the embryo development [Ref 17], the outlet reservoir was connected to a syringe pump to apply a steady flow field and to control the flow rate. In addition, it’s for another purpose for retrieving the cultured embryos from the outlet reservoir. We’ve addressed clearly in revised manuscript.
- The quality of Figure 4 should be improved. Fluorescence image after labeling would help to express their results.
Answer:
Yes, this is a great concern. Because this is a continuous microfluidic structure chip, all the processes are completed and recorded in the video imaging system, then screengrab an appropriate image for further analysis. We’ve revised a clearer image with appropriate marking in the revised manuscript.
- The trapping efficiency of sperm and oocyte should be addressed.
Answer:
Since only a single sperm can successfully fertilize an egg under natural fertilization, therefore we did not consider the role of trapping efficiency in this study. In addition, there is no sperm flowing away when the sperm go through the electrode, most of the sperm will be grasped by the electrode and then fertilize naturally with oocyte. We use the fertilization rate and development rate as the chip's effectiveness and representative the success rate.
- The shape of the sperm should be taken into account to estimate transmembrane potential.
Answer:
Yes, this is a great comment. Human sperm morphology is the one of the most important indicators to judge sperm function and quality. In this study, we used a murine in vitro gametes model, the rodent's sperm head is not conical like a human sperm. They are flat and carry small hooks. This unique structure of sperm allows them to hook their heads together to enhance reproduction. This is a great suggestion and careful consideration should be given to future human experiments.

Reviewer 2 Report
Using a Dielectrophoretic Microfluidic BioChip Enhanced Fertilization of Mouse Embryo invitro
The paper presents a novel idea of using microfluidic system imitating an oviduct to achieve invitro fertilization using dielectrophoretic trapping. Few questions need to be answered for this manuscript.
- Main concern in this manuscript is that there is no discussion on what the advantages are for using this system. Figure 5A presents a graph with % fertility rate for DEP device and traditional IVF. The graph does not show any significant difference between these techniques. Authors need to highlight what advantages their technique can have over traditional technique.
- Authors also need to discuss how their technique can be developed further for aiding the oligozoospermia patients and what would be the challenges and limitations of this approach.
- In section 2.3, the evidence that the voltage does not exceed transmembrane potential justifies that the cells will not get electroporated, however exposure to electric field can cause changes in cell DNA. The damage caused in the cells also depends on experimental time. The authors need to provide evidence that the technique used here does not modify the cell DNA. For example, see the reference below for effect of short-term exposure on neural cells:
Lu, J.; Barrios, C.A.; Dickson, A.R.; Nourse, J.L.; Lee, A.P.; Flanagan, L.A. Advancing practical usage of microtechnology: A study of the functional consequences of dielectrophoresis on neural stem cells. Integr. Biol. 2012, 4, 1223
- The calculation used for cell membrane potential used spherical particle, however the sperm is not spherical. Authors need to provide further evidence to support this information or validate the cell viability at the electric field exposure for 1 min.
- In Figure 3A, the number of samples used for testing with DEP buffer is almost 40% of the sample size for the other tests. Why is there a discrepancy? Due to such a large difference in sample sizes, the comparison between fertility rates seems unconvincing.
- In Figure 3C, based on the images, it seems that the development of the zygote in the DEP Buffer is different from the HTF and KSOM. For example, the images for Morula and Blastocyst with DEP Buffer look different than the other two cases. Please justify the reason for such a difference.
- In Figure 4C, it is evident that the oocytes are trapped on the microfluidic structure after the DEP is switched off, however based on the image, it is not clear how the authors are able to contain the sperms which are much smaller than the oocytes. If the sperms are lost from the gaps between the structures, the previous step of concentrating the sperms will lose its effect.
- In section 2.3, the authors have specified the values of permittivity and conductivity used for different cells, please provide the references for these values.
- The authors have connected the syringe pump to the outlet unlike in traditional microfluidic devices where the syringe pump is used for pushing the liquid at the inlet. Is there any specific reason for choosing this approach?
- Is there any particular reason for using the flow rates stated in the manuscript? In some instances, after DEP is switched off, the cells remain adhered to the electrodes and higher flow rates are helpful to push these cells away from electrodes. Did you see any such phenomena? Do higher flow rates affect the device performance?
- Please provide details about the composition of the DEP buffer used here. Providing brief background theory about DEP will also be helpful.
- Please proof read the manuscript for grammatical and formatting errors. The manuscript organization needs to be improved in some places.
Author Response
Reviewer 2
Comments and Suggestions for Authors
Using a Dielectrophoretic Microfluidic BioChip Enhanced Fertilization of Mouse Embryo in vitro.
The paper presents a novel idea of using microfluidic system imitating an oviduct to achieve invitro fertilization using dielectrophoretic trapping. Few questions need to be answered for this manuscript.
- Main concern in this manuscript is that there is no discussion on what the advantages are for using this system. Figure 5A presents a graph with % fertility rate for DEP device and traditional IVF. The graph does not show any significant difference between these techniques. Authors need to highlight what advantages their technique can have over traditional technique.
Answer:
In this study, the rate of fertilization is proportional to the sperm concentration within the range of 18,000-240,000 sperm counts in the DEP microfluidic chip and traditional IVF group. The rate of fertility in vitro in DEP microfluidic chip not significantly increased 6.9 % at the sperm 9000 and 21.1 % at sperm 3000 beyond that with the traditional IVF group. At total sperm 3000, the fertility rate was improved 21.1 % in our DEP microfluidic chip. We’ve demonstrated that even a small number of sperm, the results show that this designed microfluidic chip is useful to provides an alternative to intracytoplasmic sperm injection for fertilization in patients with severe oligospermia. We’ve revised and highlight the advantage in revised manuscript.
- Authors also need to discuss how their technique can be developed further for aiding the oligozoospermia patients and what would be the challenges and limitations of this approach.
Answer:
Yes, this is also the main basis for our development of this chip, because oligozoospermia patients cannot fertilize in nature traditionally in IVF program, it must be carried by ICSI. Our chip is focused on hoping to replace invasive ICSI here. We’ve verified in the manuscript according to reviewer’s suggestion.
- In section 2.3, the evidence that the voltage does not exceed transmembrane potential justifies that the cells will not get electroporated, however exposure to electric field can cause changes in cell DNA. The damage caused in the cells also depends on experimental time. The authors need to provide evidence that the technique used here does not modify the cell DNA. For example, see the reference below for effect of short-term exposure on neural cells:
Lu, J.; Barrios, C.A.; Dickson, A.R.; Nourse, J.L.; Lee, A.P.; Flanagan, L.A. Advancing practical usage of microtechnology: A study of the functional consequences of dielectrophoresis on neural stem cells. Integr. Biol. 2012, 4, 1223
Answer:
Yes, this is a great comment. Some concerns are addressed on the effect of an electric field system on embryogenesis. The application of electric fields at small frequency to cells has a long and contentious history. All work indicates that developing embryos produce endogenous currents involved during embryonic development. Electric activation can facilitate the fertilization and early embryonic development after ICSI (Intracytoplasmic Sperm Injection) in human beings. As electric stimulation is known to enhance protein and DNA synthesis, to stimulate neuronal cells and to differentiate the migration of neural crest cells. Our previous work might provide some explanation of the dynamic cultural result relative to a static cultural result that proved a dynamic culture in the electro-wetting microfluidic system to enhance the cleavage and development of a mouse embryo and to accelerate the growth to the blastocyst stage (# 20). As little is known about the sensitivity to electric fields of mammalian embryos of various species and at various stages of the early preimplantation development, further investigation is necessary. We’ve verified in the revised manuscript with additional reference according the reviewer’s suggestion.
- The calculation used for cell membrane potential used spherical particle, however the sperm is not spherical. Authors need to provide further evidence to support this information or validate the cell viability at the electric field exposure for 1 min.
Answer:
Yes, this is a great comment. In this study, we used a murine in vitro gametes model, the rodent's sperm head is not conical like a human sperm. They are flat and carry small hooks. This unique structure of sperm allows them to hook their heads together like a cell mass to enhance reproduction. This kind of sperm cell mass might mimic a spherical particle. This is a great suggestion and careful consideration should be given to future human experiments.
- In Figure 3A, the number of samples used for testing with DEP buffer is almost 40% of the sample size for the other tests. Why is there a discrepancy? Due to such a large difference in sample sizes, the comparison between fertility rates seems unconvincing.
Answer:
Yes, this is great concern. Since this is a mammalian murine model. The biological variability in different patch of animal experiments was possible and there is no significant statistical difference between DEP buffer solution and traditional culture medium. Most importantly, the result indicates that the rate of fertility in DEP microfluidic device is better than with traditional droplet- based IVF at very low sperm concentration.
- In Figure 3C, based on the images, it seems that the development of the zygote in the DEP Buffer is different from the HTF and KSOM. For example, the images for Morula and Blastocyst with DEP Buffer look different than the other two cases. Please justify the reason for such a difference.
Answer:
Yes, in Figure 3C, the morula stage of embryo in DEP group developed a little bit slower and not yet fully compact. But this is biological variability within the allowable range. Frankly we can't explain and it should also have nothing to do with different groups of experiments.
- In Figure 4C, it is evident that the oocytes are trapped on the microfluidic structure after the DEP is switched off, however based on the image, it is not clear how the authors are able to contain the sperms which are much smaller than the oocytes. If the sperms are lost from the gaps between the structures, the previous step of concentrating the sperms will lose its effect.
Answer:
The concept of the DEP microfluidic chip our experiment is that the mouse oocytes and sperm become trapped on the electrodes with the p-DEP, with further flow to be co-incubated at the micro-structures for natural insemination (in the Chip portion C). The pocket structure is for the purpose of observation of appearance of pronuclei formation, or unfertilized oocytes after switched off the DEP. We’ve revised in revised manuscript.
- In section 2.3, the authors have specified the values of permittivity and conductivity used for different cells, please provide the references for these values.
Answer:
We’ve revised additional references for the values of permittivity and conductivity in this study (Ref #18 & 19).
- The authors have connected the syringe pump to the outlet unlike in traditional microfluidic devices where the syringe pump is used for pushing the liquid at the inlet. Is there any specific reason for choosing this approach?
Answer:
Yes, this is a great review. The experiment was observed with an optical microscope and the electrode pads of the DEP microfluidic chip were connected into a function generator to regulate the AC voltage and frequency. The flow rate and the method of flow delivery might be important factors to affect the embryo development [Ref 17], the outlet reservoir was connected to a syringe pump to apply a steady flow field and to control the flow rate. In addition, it’s for another purpose for retrieving the cultured embryos from the outlet reservoir.
- Is there any particular reason for using the flow rates stated in the manuscript? In some instances, after DEP is switched off, the cells remain adhered to the electrodes and higher flow rates are helpful to push these cells away from electrodes. Did you see any such phenomena? Do higher flow rates affect the device performance?
Answer:
The concept of the DEP microfluidic chip our experiment is that the mouse oocytes and sperm become trapped on the electrodes with the p-DEP, with further flow to be co-incubated at the micro-structures for natural insemination (in the Chip portion C). The pocket structure is for the purpose of observation of appearance of pronuclei formation, or unfertilized oocytes after switched off the DEP. We’ve revised in revised manuscript.
- Please provide details about the composition of the DEP buffer used here. Providing brief background theory about DEP will also be helpful.
Answer:
The composition of DEP buffer consisted of an aqueous solution of sugar with the following percentage: 9.5% sucrose (S7903, Sigma-Aldrich), 0.1 mg/ml dextrose (D9559, Sigma-Aldrich), 0,1% pluronic F68 (Pluronic F68 non-ionic surfactant 100 ×, Gibco). We’ve revised this information in revised manuscript.
- Please proof read the manuscript for grammatical and formatting errors. The manuscript organization needs to be improved in some places.
Answer:
We’ve revised and improved by English expert according to reviewer’s suggestion.

Round 2
Reviewer 1 Report
Although the authors resubmit modified manuscript, merit of the proposed microfluidic device is not clear. The authors argue that they achieve results with a very low concentrations of sperm but traditional IVF methods also allow us to achieve similar results as shown in Fig. 5. I cannot agree that fertility ratio of 17.2 ± 7.5 % by microfluidic device is better than that of 14.2 ± 7.5 % by traditional IVF method. The data show large standard deviation, and mean value is quite similar between microfluidic device and traditional method. If increase of sperm concentration by DEP is effective for the fertilization, slightly higher concentrations of sperms have to show higher fertilization ratio until the fertilization ratio become saturation. Hence, I cannot agree that the proposed microfluidic device enhance fertilization of mouse embryo compared with traditional IVF methods. I recommend rejection of the manuscript.
Reviewer 2 Report
Authors have answered the questions satisfactorily!